# The effect of notification window length on the epidemiological impact of COVID-19 contact tracing mobile applications

Trystan Leng [1,2,3✉], Edward M. Hill [1,2], Matt J. Keeling [1,2], Michael J. Tildesley [1,2,4] & Robin N. Thompson [1,2,4]

## Abstract

**Background** The reduction in SARS-CoV-2 transmission facilitated by mobile contact tracing applications (apps) depends both on the proportion of relevant contacts notified and on the probability that those contacts quarantine after notification. The proportion of relevant contacts notified depends upon the number of days preceding an infector's positive test that their contacts are notified, which we refer to as an app's *notification window*.

**Methods** We use an epidemiological model of SARS-CoV-2 transmission that captures the profile of infection to consider the trade-off between notification window length and active app use. We focus on 5-day and 2-day windows, the notification windows of the NHS COVID-19 app in England and Wales before and after 2nd August 2021, respectively.

**Results** Our analyses show that at the same level of active app use, 5-day windows result in larger reductions in transmission than 2-day windows. However, short notification windows can be more effective at reducing transmission if they are associated with higher levels of active app use and adherence to isolation upon notification.

**Conclusions** Our results demonstrate the importance of understanding adherence to interventions when setting notification windows for COVID-19 contact tracing apps.

## Plain language summary

After submitting a positive SARS-CoV-2 test result, mobile contact-tracing apps identify 'recent' high-risk encounters with other app users, who are then notified of potential exposure. An app's success at limiting further transmission depends on the proportion of infected contacts notified. This depends on what counts as 'recent', e.g. notifying contacts from 5 days prior to the positive test can capture more infections than notifying contacts from 2 days prior. We call this number of days an app's *notification window*. However, an app's effectiveness also depends on whether or not exposed contacts use the app and adhere to isolation if notified. If shorter windows are associated with higher levels of active app use, they can be more effective at reducing transmission than longer windows, demonstrating the importance of considering the potential impact on active app use when setting an app's notification window length.

[1] The Zeeman Institute for Systems Biology & Infectious Disease Epidemiology Research, School of Life Sciences and Mathematics Institute, University of Warwick, Coventry, UK. [2] JUNIPER—Joint UNIversities Pandemic and Epidemiological Research, https://maths.org/juniper/. [3] Department of Infectious Disease Epidemiology, Imperial College London, London, UK. [4] These authors contributed equally: Michael J. Tildesley, Robin N. Thompson. ✉email: trystan.leng@imperial.ac.uk

The automated tracing of close contacts via mobile phone applications (apps) has been used in many countries to reduce SARS-CoV-2 transmission[1]. In England and Wales, the National Health Service (NHS) COVID-19 contact tracing app has been available since 24th September 2020[2]. After a user submits a positive test result, apps identify via bluetooth the user's recent high-risk encounters with other app users based on a number of factors, such as proximity and duration of contact. For the NHS COVID-19 app, a high-risk encounter is defined as being within two metres of someone for at least fifteen minutes[3], though their risk scoring algorithm also considers the app user's likely infectiousness on the day of encounter. If the user is symptomatic when entering their positive test result in the app, high-risk encounters $w$ days prior to symptom onset up until the present moment are identified, while if a user is asymptomatic, high-risk encounters $w$ days prior to the individual taking the positive test up until the present moment are identified. Contacts identified as involved in a high-risk encounter are then notified of potential exposure. We refer to $w$ as an app's *notification window*.

An app's effectiveness at reducing transmission depends on the proportion of an infected individual's contacts who are identified through the app. It might therefore be expected that longer notification windows will lead to greater reductions in transmission. However, long notification windows may have negative consequences, such as a large number of notifications being issued to uninfected individuals, with potential impacts on app usage. In England, over one million notifications were sent to contacts in the first 2 weeks of July 2021[4], leading some commentators to suggest that many individuals identified through the app were isolating unnecessarily. On 2nd August 2021, the notification window was reduced from 5 days to 2 days for asymptomatic individuals submitting a positive result[5], to encourage the continued use of the NHS COVID-19 app while limiting the number of uninfected individuals isolating.

If stronger measures are necessary to mitigate SARS-CoV-2 transmission in the future, increasing the notification window would seem an intuitive response because the number of potential infectious contacts notified would increase. However, the effectiveness of contact tracing not only depends on the proportion of infected individuals notified, but also depends on people's behaviour upon notification[6]. In part, an app's effectiveness depends upon the likelihood that an infected individual's contacts actively use the app and adhere to the recommended self-isolation period, which in turn depends on the perceived risk that a notified individual is infected. As well as increasing the proportion of infectious contacts notified, longer notification windows increase the number of uninfected individuals asked to self-isolate, which may reduce public confidence and lead to lower levels of active app use.

Here, we analyse the expected number of primary cases infected by a base case who reports their infection to a contact tracing app. We then consider the expected number of secondary cases infected by those primary cases (illustrated in Fig. 1), and explore the effectiveness of app-based notifications at reducing transmission with either a 2-day or a 5-day notification window at different levels of active app use. Rather than aiming to generate precise quantitative predictions, our goal is to use a simple epidemiological model to explore the general impacts of different notification windows on SARS-CoV-2 transmission. Using this approach we find that, if the same level of active app use is assumed, 5-day windows result in larger reductions in transmission than 2-day windows. However, if 2-day windows are associated with higher levels of active app use than 5-day windows, they can become more effective at reducing transmission.

## Methods

In our epidemiological model, a base case app user who becomes infected on day 0 and who tests positive to SARS-CoV-2 on day $d$ is considered (for details about the model in addition to those explained in the main text, see Supplementary Methods). To explore the effect of notification window length on transmission in a concrete setting, we consider an asymptomatic base case who is detected using a lateral flow test (LFT), with no delay between taking a test and receiving a positive test result (for a separate analysis in which an asymptomatic base case is assumed to seek a PCR test, with a two day delay, see Supplementary Note 1 and Supplementary Fig. 1; for an analysis in which the base case is symptomatic and detected at symptom onset, see Supplementary Note 2 and Supplementary Fig. 2). The relative probability of a base case testing positive on a given day $d$ varies through time (Fig. 2a), which we obtained by normalising a previously published test probability profile for LFTs[7]. This is equivalent to assuming that the base case tests once at a random time during infection, and is detected - the impact of regular testing is considered in Supplementary Note 3 and Supplementary Figs. 3–4. We assume that the base case self-isolates after taking a test, and that self-isolation is perfectly effective. The expected number of primary infections from the base case prior to taking a test is informed by a previously derived SARS-CoV-2 infectiousness profile[8], under the assumption that contacts occur randomly at a constant rate until taking a test. Primary cases infected within the notification window (grey shaded area in Fig. 1a) are notified of possible exposure, while those infected before the notification window receive no notification. Primary cases infected within the notification window self-isolate with probability $p$, with $p$ representing the proportion of the population who are active app users. We define active app use as both having the app (downloaded and with bluetooth enabled) and adhering to isolation upon notification. Those who are infected before the notification window or are not active app users continue mixing with the population throughout their infectious period if asymptomatic, or until symptom onset if symptomatic, at which point they self-isolate. We assume that asymptomatic cases comprise 30% of all cases[9] and are 50% as infectious as symptomatic cases[10].

In our model, individuals are either not vaccinated or are fully vaccinated. We assume that 70% of the population are fully vaccinated, in line with estimates of vaccine uptake in August 2021 among the adult population in the UK. Based on estimates averaging over multiple vaccine products during the Delta wave, we assume that vaccination reduces susceptibility to infection by 63%[11] and transmissibility upon infection by 63%[12]. We assume that vaccinated primary cases do not self-isolate upon notification, as they have not been legally required to self-isolate upon notification since 16th August 2021[13]. The impact of vaccine efficacy on our results is explored in Supplementary Note 4 and Supplementary Fig. 5. We use a previously derived incubation period distribution[14] to determine the time from infection to symptom onset for symptomatic cases. To estimate the number of infections that each primary case is expected to generate, we directly calculate the effective reproduction number, $R^*$, as the ratio between the expected number of secondary infections and the expected number of primary infections (Fig. 1d). Considering the expected number of secondary cases arising from primary cases is essential to quantify the impacts of different notification windows, as the expected number of primary cases is not affected by contact tracing (specifically, the expected number of primary cases depends only on when the base case isolates, and not on the notification window; our focus is the number of onwards transmissions prevented from primary cases as a result of the choice of notification window).

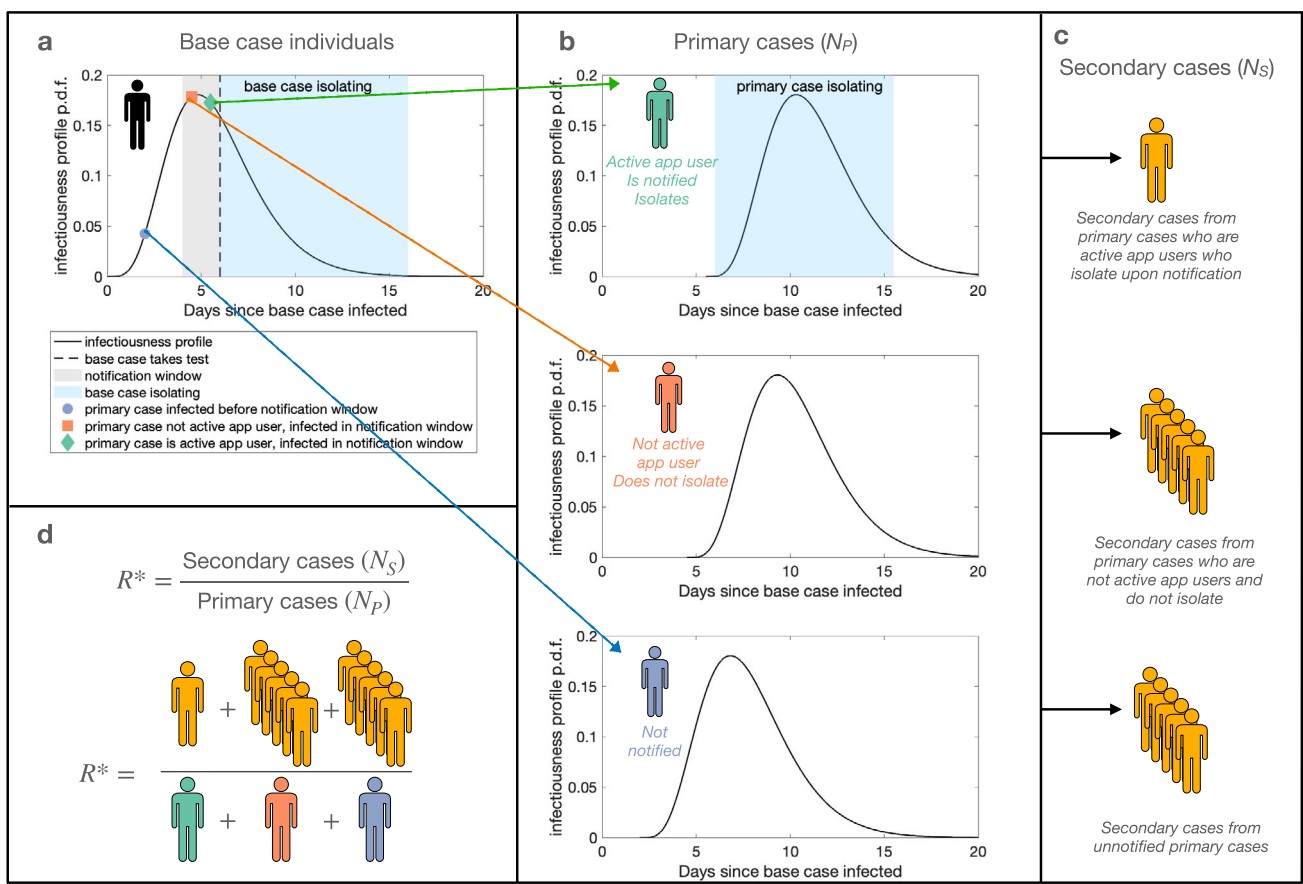

**Fig. 1 Schematic illustrating the modelling approach. a** The infectiousness of a base case individual varies through time. A base case individual takes a test on day $d$ in their infectiousness profile (dashed line), after which they isolate (light blue shaded area). **b** Primary cases with the app who are infected by the base case from day $d-w$ to day $d$ are notified of possible exposure. A proportion $p$ of individuals infected within the notification window adhere to self-isolation (green, top panel), while a proportion $(1-p)$ of individuals are not active app users and do not self-isolate (orange, middle panel). Adhering individuals self-isolate from notification until $i_{notif}$ days have elapsed since contact with the base case. Primary cases infected before day $d-w$ are not notified (blue, bottom panel). Those who do not adhere to isolation upon notification or who are not notified either mix normally in the population throughout their infectious period or they isolate upon symptom onset. **c** The expected number of secondary cases resulting from non-adhering or unnotified primary cases is higher than the expected number of secondary cases that result from primary cases who adhere to isolation after notification. **d** $R^*$ is calculated as the ratio between the expected number of secondary cases and the expected number of primary cases—in our illustrative example, the expected number of secondary cases is 11, and the expected number of primary cases is 3, giving $R^* = 11/3$.

**Reporting summary**. Further information on research design is available in the Nature Research Reporting Summary linked to this article.

## Results

We first considered the impact of the notification window length on transmission, assuming that all primary cases are active app users (i.e. all primary cases infected within the notification window adhere to self-isolation upon notification). The reduction in $R^*$ resulting from app-based notifications (relative to a scenario in which a contact tracing app is not used) increases with the notification window length (Fig. 2b), though there are only limited further benefits in transmission reduction for notification windows longer than 5 days. Assuming 100% active app use, then under the baseline parameter values considered here a 2-day notification window results in a 32% reduction in $R^*$ while a 5-day notification window results in a 52% reduction in $R^*$. Even long notification windows do not eliminate transmission entirely, as primary cases may transmit the virus before they are notified of their contact with the base case.

Next, considering more realistic scenarios in which not all individuals are active app users, we calculated the reduction in $R^*$

for different levels of active app use. For a given notification window duration, the reduction in $R^*$ increases linearly with the proportion of primary cases who actively use the app and adhere to isolation (Fig. 2c). This is because, if the proportion of active app users is doubled, then the expected number of primary cases whose infectious period will be reduced due to app-based notification (and isolation) will also be doubled. A higher level of active app use is required to obtain a given reduction in $R^*$ for a 2-day window than for a 5-day window. For example, with a 5-day window, 58% of primary cases must be active app users to reduce $R^*$ by 30%, while a 2-day window requires 95% of primary cases to be active app users to obtain the same 30% reduction in $R^*$. If the proportion of people who actively use the app is high (above 60%), then a 5-day window is always more effective than a 2-day window, irrespective of the level of active app use with a 2-day window.

Assuming the same level of active app use, longer notification windows result in larger reductions in transmission. However, if fewer individuals are active app users when a longer notification window is chosen, then a 2-day window can lead to a greater reduction in transmission than a 5-day window (Fig. 2d). Under the model considered here and its baseline parameterisation,

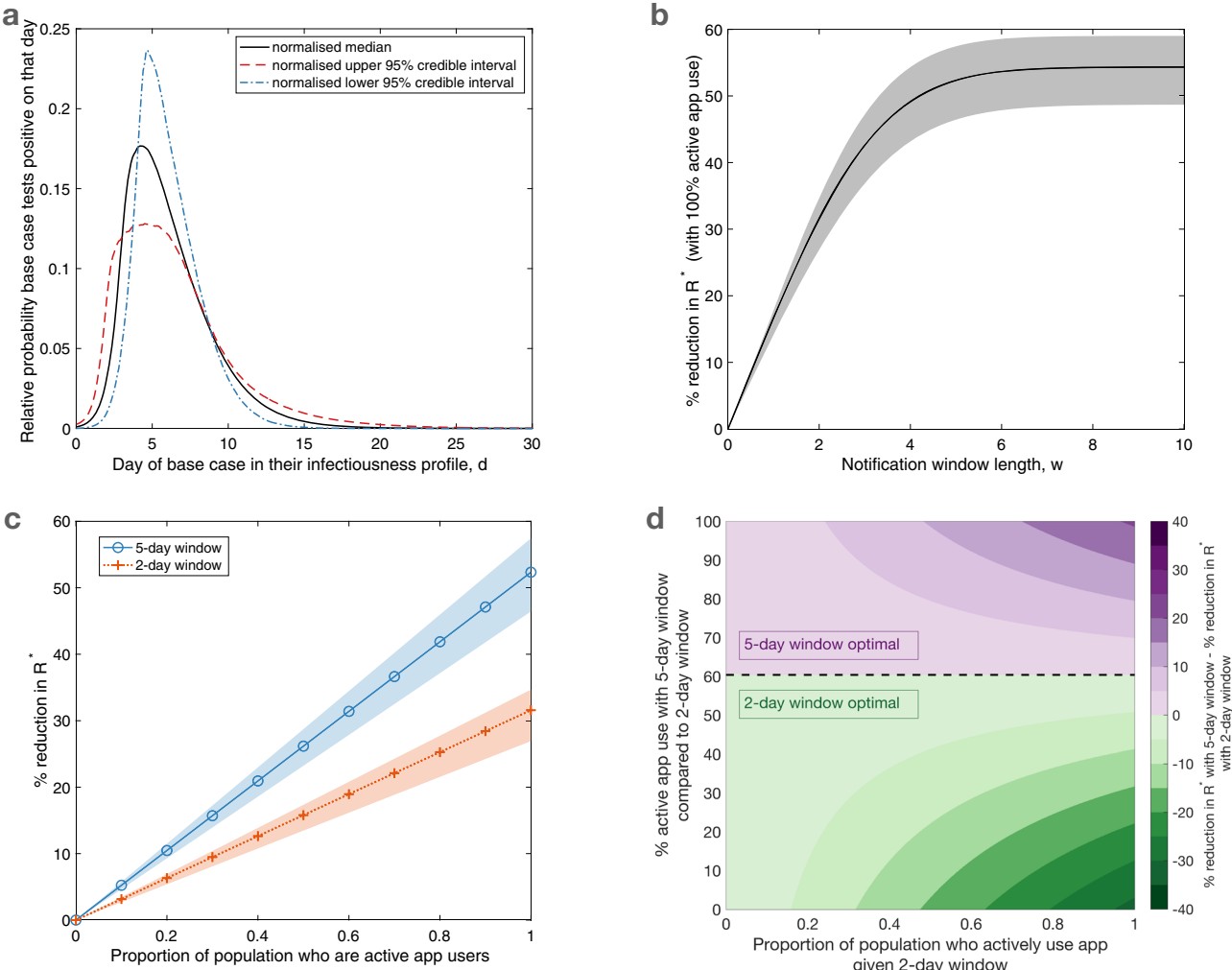

**Fig. 2 Impact of the notification window length and active app use on transmission. a** The relative probability of the base case testing positive on a given day in their infectiousness profile, obtained by normalising the median (black, solid line) positive test probability profile for LFTs taken by asymptomatic, infected individuals[7]. Normalised 95% credible interval test probability profiles[7] (upper - red, dashed line; lower - blue, dot-dashed) are also considered, to obtain shaded regions in (**b**) and (**c**). **b** The percentage reduction in $R^*$ for different length notification window $w$, relative to a scenario in which a notification app is not used, under the assumption that all individuals are active app users (i.e. 100% adherence). **c** The relationship between the proportion of primary cases who are active app users and the percentage reduction in $R^*$ for a 5-day notification window (blue solid line, circle markers) and a 2-day notification window (orange dotted line, cross markers). **d** A heat map indicating the transmission reduction achieved by using a 5-day window rather than a 2-day window, quantified by the difference in the percentage reduction in $R^*$ that results from a 5-day notification window compared to a 2-day notification window. The proportion of primary cases assumed to be active app users for a 2-day window is shown on the x-axis, and the relative level of active app use assumed for a 5-day window (compared to the level of active app use for a 2-day window) is shown on the y-axis. Purple (green) regions correspond to where 5-day (2-day) notification windows lead to a larger reduction in $R^*$.

when the level of active app use for a 5-day window is less than 60% of the level of active app use for a 2-day window, the 2-day window results in less transmission. In other words, if more than four out of ten individuals who would actively use the app under a 2-day window would no longer actively use the app under a 5-day window, then the increase in secondary cases from individuals no longer actively using the app is greater than the reduction in secondary cases that results from capturing more primary cases in the notification window.

## Discussion

Contact tracing mobile apps have been used to reduce SARS-CoV-2 transmission in countries worldwide during the COVID-19 pandemic. Key questions for policy makers when these apps are used include how to define a close contact, and how long the

notification window should be. In this research, we have considered the second of these questions, and shown how mathematical modelling can be used to predict the effects on transmission of different notification window lengths. If the notification window is too short, then large numbers of infected contacts will be missed. On the other hand, large numbers of infected contacts may also be missed for long notification windows, if long notification windows are associated with low levels of active app use. If this is the case, then intermediate length notification windows may balance these two factors, reducing transmission maximally.

Adopting a parsimonious approach, our results demonstrate the complexity in assessing the optimal notification window length for mobile contact-tracing apps. A 5-day window is considerably more effective at reducing transmission than a 2-day

window if the level of active app use in the population is not affected by the notification window length. If high levels of active app use can be achieved for a 5-day window, this strategy will always be optimal. However, by 4th August 2021, there had been ~27.3 million downloads of the NHS COVID-19 app in England and Wales[4]. With an adult population of over 48 million[15], a substantial proportion of eligible users had not downloaded the app, indicating at best an intermediate level of active app use.

If low levels of active app use are associated with long notification windows, then short notification windows may lead to higher reductions in transmission than long notification windows. For the NHS COVID-19 app, there are publicly available data regarding the number of notifications for each case reported on the app[4]. The numbers of positive tests inputted into the app are also recorded, and can be combined with case data from UKHSA[16] to infer the proportion of all known positive tests that are reported on the app[17]. These data may provide an insight into the impact of app rule changes. However, these data will also be impacted by behavioural trends at the time. For example, while the number of notified individuals per case reported on the app decreased after the notification window was reduced from 5 days to 2 days for asymptomatic individuals on the 2nd August 2021, it was already decreasing prior to the rule change. The proportion of all positive tests reported on the app decreased both before and after the rule change. This trend tentatively suggests that the change in the notification window duration had a limited impact on users' likelihood of engagement with the app; however, this decrease may have occurred more rapidly without the rule change.

In our main analysis, we assumed that base cases receive a positive LFT result on a random day in their infectious period. In reality, this detection time distribution has varied throughout the course of the epidemic[18] and will be influenced by a variety of factors, such as the requirements of different workplaces regarding regular testing. Understanding this distribution is important, as the relative impact of different notification windows at different levels of active app use depends on when base cases are detected (Supplementary Note 3). Early detection not only reduces the expected number of secondary cases from infected individuals, but also increases the proportion of secondary cases captured in scenarios with shorter notification windows. If cases are detected early, the secondary cases missed by shorter windows may be offset by only modest increases in active app use.

Previous studies have demonstrated that the extent of active usage of mobile contact tracing apps has a large impact on their effectiveness[18,19]. The time to notification of exposed contacts also plays an integral role for SARS-CoV-2[20]; this is corroborated by our results regarding the contrasting reduction in $R^*$ based on whether a user inputs their positive result from an LFT or from a PCR test (where the PCR test involves a delay between the test date and the positive test result). A range of other factors may also impact the effectiveness of app-based measures. For example, a policy that increases the duration of self-isolation may increase the risk of transmission, if such a policy also leads to a decrease in the rate of self-reporting and adherence[6]. It has been suggested that contact tracing has the potential to delay epidemics, in part because of considerable heterogeneity in transmission of SARS-CoV-2 between different infected individuals[21]. While we find that heterogeneity in contact rates or duration of infection does not impact our estimates of $R^*$ (Supplementary Note 5 and Supplementary Figs. 6–7), the previously described factors nevertheless demonstrate that the impacts of heterogeneity between hosts on population-scale transmission should be explored further.

While our study considers the impact of reducing the number of notified individuals by shortening the length of the notification window, a reduction in the number of notified individuals could also be achieved by changing the definition of a high-risk encounter. While the current definition used in the UK involves being within two metres of someone for at least fifteen minutes[3], this threshold distance could be decreased or the threshold duration of contact increased. Sensitivity of our results to either of these factors could be approximated in our framework by reducing the probability of primary cases being notified via the app. Though there are relatively few studies considering the impact of mobile contact tracing apps specifically, studies exploring the impact of contact tracing more generally have considered the impact of reducing the proximity or increasing the duration of contact for someone to be regarded as a "close contact" for SARS-CoV-2. The impact of duration has been considered explicitly[22], while other studies have explored the impact of reducing the overall percentage of contacts traced[23,24], including through varying the notification window[8]. Despite practical limitations preventing precise measurements of both proximity and duration of contact through mobile devices[25], future studies exploring the effects of these factors on transmission would be a valuable line of research.

As with any epidemiological modelling study, our analyses involve a number of simplifying assumptions. We assumed that vaccinated individuals infected within the notification window do not self-isolate, in line with legal requirements since 16th August 2021[13]. However, some proportion of vaccinated primary cases may isolate upon notification, increasing the effectiveness of a contact tracing app. We treated adherence to isolation upon notification as binary— either individuals isolate for the entirety of their infectious period, or not at all. In reality, some notified individuals may be partially adherent (i.e. they may have fewer contacts than normal, but not isolate completely, or they may isolate for only part of their infectious period). Further, whether or not an individual isolates may depend on their own subjective evaluation of their risk, and they may not be aware of rule changes affecting notification through contact tracing apps. In our model, we assumed that contacts occur at random and contacts between specific individuals are not repeated, so a 2-day window will find 40% of the infected contacts that a 5-day window would find. In reality, this proportion would be expected to be higher because of repeated contacts. We assumed that the definition of a high-risk encounter remains the same throughout an individual's notification window. Some tracing apps, such as the NHS COVID-19 app, also factor in an individual's likely infectiousness on the day of encounter. The infectiousness profiles of symptomatic, asymptomatic, and vaccinated individuals were assumed to be equal in this study (at least up to the time at which symptomatic individuals develop symptoms, at which point they isolate), though the profiles of asymptomatic and vaccinated individuals were scaled so that they are expected to generate fewer infections overall. Further, we assumed that the infectiousness profile of symptomatic individuals and their incubation period are independent of each other (in other words, an individual's infectiousness was assumed to be independent of the time at which they develop symptoms). This simplifying assumption is common in the literature[20,26,27], although methods relaxing this assumption have also been explored[8,28,29], providing an avenue for further development of our research. Infectiousness profiles, incubation period distributions and detection time distributions used within the model were derived from data from the 'wild-type' strain of SARS-CoV-2. Emerging strains of SARS-CoV-2 may impact each of these distributions (e.g. Hart et al. estimate the infectiousness profile for different SARS-CoV-2

variants[30]), and may be an important factor to account for in future studies.

In summary, if levels of active contact tracing mobile app use are low when the notification window is long, and if individuals are more likely to use the app and adhere to its guidance for shorter notification windows, then the benefits of a longer notification window may be outweighed by the costs of lower active app use. When making decisions about the length of the notification window, policy makers should consider the potential impacts on active app use, and the resulting effects on transmission. We have provided a quantitative framework for guiding these decisions.

## Data availability

The raw data used in this study and the source data for the results figures are provided in the GitHub repository associated with the study: https://github.com/tsleng93/AppModelling.

## Code availability

Code for the study is available at: https://github.com/tsleng93/AppModelling Archived code at time of publication: https://doi.org/10.5281/zenodo.6482890[31].

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

## Acknowledgements

We thank Christopher Davis and Laura Guzmán Rincón for comments on an early draft, and Laura Guzmán Rincón for reviewing the model code. Thanks also to the anonymous reviewers for comments that helped us to improve the initial version of this paper.

T.L., M.J.K. and M.J.T. were supported by UKRI through the JUNIPER modelling consortium [grant number MR/V038613/1]. M.J.K., R.N.T. and M.J.T. were supported by the Engineering and Physical Sciences Research Council through the MathSys CDT [grant number EP/S022244/1]. E.M.H., M.J.K. and M.J.T. were supported by the Biotechnology and Biological Sciences Research Council [grant number: BB/S01750X/1]. R.N.T. was supported by UKRI through the Rapid Assistance in Modelling the Pandemic continuity grant [grant number: EP/V053507/1]. M.J.K. was supported by the National Institute for Health Research (NIHR) [Policy Research Programme, Mathematical & Economic Modelling for Vaccination and Immunisation Evaluation, and Emergency Response; NIHR200411]. M.J.K. is affiliated to the National Institute for Health Research Health Protection Research Unit (NIHR HPRU) in Gastrointestinal Infections at University of Liverpool in partnership with UK Health Security Agency (UKHSA), in collaboration with University of Warwick. M.J.K. is also affiliated to the National Institute for Health Research Health Protection Research Unit (NIHR HPRU) in Genomics and Enabling Data at University of Warwick in partnership with UK Health Security Agency (UKHSA). The views expressed are those of the author(s) and not necessarily those of the NHS, the NIHR, the Department of Health and Social Care or UK Health Security Agency. The funders had no role in study design, data collection and analysis, decision to publish, or preparation of the paper.

## Author contributions

R.N.T. and M.J.T. conceptualised this study. R.N.T., M.J.T and T.L. developed the methodology. T.L. wrote the model code, undertook the modelling analysis, and analysed and visualised the results. E.M.H. validated the model code and results. T.L. wrote the original draft of the paper, and T.L., E.M.H., M.J.K., M.J.T. and R.N.T. reviewed and edited the paper.

## Competing interests

The authors declare no competing interests.

**Additional information**

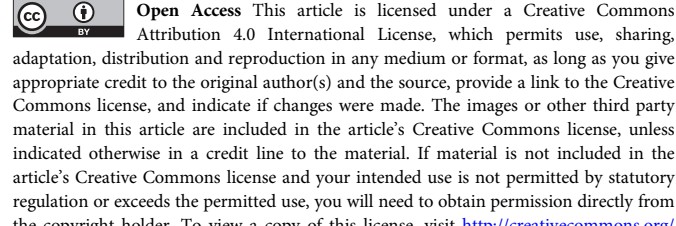

