## [Peer Review File · Communications Medicine]

Reviewers' comments:

Reviewer #1 (Remarks to the Author):

This is a solid modeling study that examined the effect of notification window length in the COVID-19 contact tracing app on the number of secondary cases infected by primary cases, quantified by the effective reproduction number R . The probabilistic model jointly considered the infectiousness profile, notification window, testing time, active app usage, and adherence to isolation upon notification. Using this model, the effectiveness of 5-day and 2-day windows was compared under assumptions of various active app-usage and adherence levels. Results indicate that a short window can be more effective with higher app usage and better adherence to isolation. This conclusion is technically sound based on the model; however, the modeling results are predicated on a number of idealized assumptions, which seem to miss several key factors affecting the transmission process. To make the study more relevant to real-world applications, a few questions and concerns need to be addressed.

1. The qualitative conclusion that a shorter notification window with higher app usage and adherence could outperform a longer window is somewhat trivial. The major contribution of the study is to identify the threshold for relative app usage (60%, as shown in Fig. 2d). But the neglect of several factors in the model may lead to inaccurate estimate of this threshold and the associated uncertainty.
2. The model did not consider the effect of vaccination. This was listed as a limitation of the study in the discussion section. As the vaccination rate in UK is currently high, neglecting the impact of vaccination on secondary cases would undermine the practical value of the results, given that vaccine can reduce population susceptibility and case transmissibility. I am not advocating performing a full exploration of all possible scenarios, which seems unfair to the authors. But I think the authors should at least incorporate the impact of vaccine in some capacity. For instance, maybe the number of secondary cases could depend on the current vaccination rate.
3. Have the authors considered the heterogeneity in individuals' infectiousness profile? Will this heterogeneity impact the threshold and the uncertainty in Fig. 2b,c? The model is based on one single primary case (with some variations in features). At the population level, there might be superspreaders and isolation may lead to different population-level prevalence. After all, the ultimate goal of using the contact tracing app is to reduce infection in the population. It might be good to see how superspreading can impact the results.
4. I am wondering if there are any data or studies showing that the app usage and adherence decrease with increased number of notifications. For each individual, the probability of effective use may not depend on the notification time window and probably is more related to the perception of risk. On the other hand, if a person is notified about an exposure occurred 5 days ago, his/her adherence may be lower than when notified about an exposure 2 days ago. In any case, having some ballpark estimates of the decrease rate could inform whether a 5-day or 2-day window is more effective.

Reviewer #2 (Remarks to the Author):

The manuscript “The effect of notification window length on the epidemiological impact of COVID-19 contact tracing mobile applications” uses a simple epidemiological model to investigate the effect of the notification window length for contact tracing apps on app usage effectiveness, in conjunction with public app usage and adherence to self-isolation. One important outcome is that, while increasing the length of the notification window allows to trace more contacts, it can be less effective of a shorter notification windows when combined with possible lower adherence.

I think the manuscript presents a highly valuable epidemiological study with important outcomes for public health, which is also strongly rooted in a concrete motivation – i.e., the change of notification policy in England and Wales in the summer 2021, and potential future decisions.

I find the paper well written, accessible to a wide readership in medical sciences and epidemiology. It presents a simple but rigorous method for evaluating effectiveness of contact tracing apps in terms of policy design (notification window) and public adherence (active app usage). The results are novel and robust, and the method looks scientifically sound.

I think one of the main messages of the paper – that a change in policy cannot be evaluated but in conjunction with the expected change in adherence – is extremely important for policy response, not only in the context of contact tracing but also for more general public health interventions.

Overall, I strongly recommend the paper for publication. I only have one main comment and several other minor comments that the authors may want to consider for a revision.

A main comment regards the use of the quantity R^* defined as the ratio between average nr of secondary cases over primary cases. On the one hand, I do agree that counting primary cases only would neglect the effective contribution to transmission from those cases, which differs substantially according to the characteristics of the primary case. On the other hand, it seems to me that presenting R^* without an absolute measure of escaped transmission (i.e., presenting the average number of secondary cases per primary case, but without quantifying the actual number of primary cases) may not completely capture the absolute amount of infectious potential that has escaped. While I would launch an open question to the authors whether a quantity like “escaped infectious potential relative to the infectious potential without tracing” (looking at primary cases only) may be an alternative reasonable measure of relative effectiveness of interventions, I would encourage the authors to expand the explanation about the reasons for the choice of R^* as currently defined. As additional minor comment, the current calculations of R^* ignore the possibility that a symptomatic primary case is tested and input the positive result into the app, leading to notification and self-isolation of secondary cases, but I think this would only introduce second order effects and would not change the main results of the paper.

Minor comments

1. On p. 1, the symbol “ d ” is used to explain the concept of notification window, however “ d ” is later used to denote the day of base case testing, and the symbol “ w ” to denote the notification window. It may be less confusing to keep using “ w ” from the beginning.

2. Caption Fig. 2, (b), sentence “under the assumption that all notified individuals adhere to isolation upon notification”: I propose to change to “under the assumption that all individuals are active app users” (i.e., they use the app and adhere upon notification)

3. The base case here is considered to be asymptomatic. I find nothing wrong with this assumption, as I expect this to be a pessimistic scenario compared to a symptomatic case who would self-isolate upon symptoms. It may however be worth to briefly mention this explicitly.

4. Supplementary material: I found the assumption of i_notif and $i_sympt = 10.5$ unusual in a continuous-time framework. Maybe mention explicitly this is an approximation, as I think that it would be possible in principle to model the “ten full days” exactly in the implementation?

5. Section S1.2: I have some doubts on some equations, which may be typos or my own mistakes, but the authors may want to check:
- Eq (9): should the lower integral bound be zero rather than “1”?
 - Two lines after (9): If isolation starts from the day of last contact (day t), should the outbound be “ $t+i_{\text{notif}}$ ” rather than “ $d+i_{\text{notif}}-t$ ” ? (with corresponding time since infection i_{notif} , as done in eq (8) ?) In Eq (10), this would result in the upper bound of the second integral to be “ i_{notif} ” – which is the time since infection after isolating for i_{notif} days from last contact – rather than “ $d-t+i_{\text{notif}}$ ”
 - Same doubts I have for the subcase (d) and corresponding equation (11)
 - All the previous points – if I haven’t made mistakes myself – should affect (13) and the implementation of the codes
 - Eq (14) there is an extra comma in the subscript
6. S1.3, first line “the ratio between the expected number of primary cases and the expected number of secondary cases”: I think the sentence should be reversed
7. S1.4, first line: change “that” to “when”
8. Eq (17) and (18): You may want to use the inequality “ $>$ ” rather than equality?

Reviewer #3 (Remarks to the Author):

I reviewed this manuscript from an epidemiological point of view. I am not a mathematical modeller.

The authors explain notification windows for asymptomatic and symptomatic base cases (which are defined differently), and state that the UK policy change from 5 days to 2 days in response to the “pingdemic” only applied to asymptomatic cases. They included both asymptomatic and symptomatic base cases in their calculations (assuming that 30% of base cases were asymptomatic), but it was not clear to me whether they applied the notification window shift from 5 days to 2 days to asymptomatics only or also to symptomatics.

The authors say that the success of contact-tracing apps depends on the proportion of infectious contacts notified (or at least I think that this is what they mean?) and the probability that those contacts quarantine after notification. However, instead of referring to the proportion of infectious contacts, they say number of contacts, number of infected individuals, number of infectious contacts, etc. in various places in the manuscript. I think what matters here is the proportion of all contacts, not the absolute numbers. Related to this, the authors assume that the contacts occur randomly at a constant rate until taking a test. I could not find any information on the assumed rate, and whether that rate was assumed to be equal for each base case. Contact matrices show that the contact rate differs substantially by age group and other factors. I realise that the authors had to simplify things but the reader needs to know what exactly they assumed regarding contact rates.

I also wondered what the authors assumed regarding the sensitivity of the lateral flow test. One of the conclusions in the supplement is that R is less reduced by PCR testing than by lateral flow testing because of the delay in receiving a test result, but people are supposed to quarantine until they receive a test result. Furthermore, lateral flow tests produce many false-negative results, especially in asymptomatics (most studies find sensitivities of only 50-60%) and therefore miss many more cases than PCR tests. Again, I realise that this may be too much detail for this paper but the authors should acknowledge this problem with their simplifications and refrain from drawing conclusions that are misleading.

The authors do not compare and contrast their findings with those by other mathematical modelers. They also do not compare their calculations with real-life data from the UK before and after the policy change on 2 August 2021. These comparisons are important precisely because their calculations are simplified versions of reality.

Finally, I was wondering whether another way out of the “pingdemic” could have been to change the definition of close contact from 2 meters to 1.5 meters (as is the case in many other countries).

Response to Reviewers: The effect of notification window length on the epidemiological impact of COVID-19 contact tracing mobile applications

We thank each reviewer for their valuable and considered comments. As you will see, we have undertaken novel analyses and made changes to the main text to address the points raised. In particular, we have included the vaccination explicitly into the modelling framework, we have considered the impact of heterogeneity in a supplementary analysis, and we have compared our findings to other models and relevant data. We believe we have fully addressed the comments of each reviewer, which we expand upon in turn below.

Reviewers' comments are in blue text while our responses are in black text. Sections of added text are in black italicised text, and the line numbers where they appear in the tracked changes manuscript are included.

Reviewer 1

This is a solid modeling study that examined the effect of notification window length in the COVID-19 contact tracing app on the number of secondary cases infected by primary cases, quantified by the effective reproduction number R . The probabilistic model jointly considered the infectiousness profile, notification window, testing time, active app usage, and adherence to isolation upon notification. Using this model, the effectiveness of 5-day and 2-day windows was compared under assumptions of various active app-usage and adherence levels. Results indicate that a short window can be more effective with higher app usage and better adherence to isolation. This conclusion is technically sound based on the model; however, the modeling results are predicated on a number of idealized assumptions, which seem to miss several key factors affecting the transmission process. To make the study more relevant to real-world applications, a few questions and concerns need to be addressed.

1. **Comment:** The qualitative conclusion that a shorter notification window with higher app usage and adherence could outperform a longer window is somewhat trivial. The major contribution of the study is to identify the threshold for relative app usage (60%, as shown in Fig. 2d). But the neglect of several factors in the model may lead to inaccurate estimate of this threshold and the associated uncertainty.

Response: In this manuscript we hope to provide a method to quantify the effectiveness of different notification window lengths at different levels of active app use. We address the specific responses mentioned by the reviewer in comments 2 and 3 below; specifically, we have added in another layer of realism by incorporating vaccinated individuals explicitly, and we have explored the impact of heterogeneity in infectiousness between individuals. However, we also agree that several factors may impact the specified threshold between a 2-day and 5-day notification window. We have expanded upon the limitations and areas for further research, which are beyond the scope of this study, such as the incorporation of more complex adherence behaviours, shorter generation times and altered infectiousness profiles for new variants, and non-modelled aspects of risk in an app's risk scoring algorithm.

Added text (lines 156-165): *“We assume that vaccinated individuals infected within the notification window do not self-isolate, in line with their legal requirements since 16th August 2021 (GOV.UK, 2021). However, some proportion of vaccinated primary cases may isolate upon notification, increasing the effectiveness of a contact tracing app. We treat adherence to isolation upon notification as binary - either individuals isolate for the entirety of their infectious period or not at all. In reality, some individuals may be partially adherent. Further, whether an individual isolates may depend on their own subjective evaluation of their risk, and may not be aware of rule changes affecting notification through contact tracing apps.”*

Further added text (lines 168-174): *“ We assume that the definition of a high-risk encounter remains the same throughout an individual's notification window. Some tracing apps, such as the NHS COVID-19 app, may also factor in an individual's likely infectiousness on the day of encounter. The infectiousness profiles of symptomatic, asymptomatic, and vaccinated individuals are assumed to be equal (at least up to the time at which symptomatic individuals develop symptoms, at which point they isolate).”*

Further added text (lines 178-181): *“Infectiousness profiles, incubation period distributions and detection time distributions used within the model were derived from data from the `wild-type' strain of SARS-CoV-2. Emerging strains of SARS-CoV-2 may impact each of these distributions (e.g. Hart et al. (2022) estimate the infectiousness profile for different SARS-CoV-2 variants), and may be an important factor to account for in future studies.”*

2. **Comment:** The model did not consider the effect of vaccination. This was listed as a limitation of the study in the discussion section. As the vaccination rate in UK is currently high, neglecting the impact of vaccination on secondary cases would undermine the practical value of the results, given that vaccine can reduce population susceptibility and case transmissibility. I am not advocating performing a full exploration of all possible scenarios, which seems unfair to the authors. But I think the authors should at least incorporate the impact of vaccine in some capacity. For instance, maybe the number of secondary cases could depend on the current vaccination rate.

Response: We have now explicitly accounted for vaccinated individuals. To do so, we make the assumption that vaccinated individuals have both reduced susceptibility and transmissibility. Assuming there is still some risk of transmission to vaccinated individuals, their inclusion will impact both primary and secondary cases. Vaccinated primary cases are not expected to isolate, in line with guidance in England as of August 2021. We now consider the impact of vaccination at different levels of vaccine efficacy in Supplement S5. The relative effectiveness of a 2-day and 5-day window is not impacted by the inclusion of vaccination, though in absolute terms contact tracing is now less effective for all notification window lengths (because vaccinated individuals are not expected to isolate upon notification)

3. **Comment:** Have the authors considered the heterogeneity in individuals' infectiousness profile? Will this heterogeneity impact the threshold and the uncertainty in Fig. 2b,c? The model is based on one single primary case (with some variations in features). At the population level, there might be superspreaders and isolation may lead to different population-level prevalence. After all, the ultimate goal of using the contact tracing app is to reduce infection in the population. It might be good to see how superspreading can impact the results.

Response: While our results are based on one infectiousness profile, this can be interpreted as the 'average' infectiousness profile of individuals within the population, and hence in some sense the method already captures heterogeneity between individuals.

To explore heterogeneity more explicitly, we simulate the model numerically (now described in Supplement S1.5 and in Algorithm 1). We explore two types of heterogeneity (Supplement S6). Firstly, reflecting heterogeneous contact rates between individuals, we explored the impact of allowing the expected number of onward cases from individuals to differ, while keeping the infectiousness profile of individuals through time the same for each individual. Secondly, we explored the impact of heterogeneity in infectious periods - individuals were assumed to have a constant level of infectiousness throughout their period of infection, and infectious periods were drawn to match the infectiousness profile at a population level. In both instances, results from numerical simulations match closely with results generated from the analytic model, as we would expect.

While the effectiveness of apps at reducing R is not impacted by heterogeneity, we now comment upon the fact that heterogeneity in contact rates and individuals infectiousness profiles may significantly delay epidemics at the population level (Gardner and Kilpatrick, 2021)

Added text (lines 132-136): *"It has been suggested that contact tracing has the potential to delay epidemics, in part because of significant heterogeneity in transmission of SARS-CoV-2 between different infected individuals (Gardner and Kilpatrick, 2021). While we find that heterogeneity in contact rates or duration of infection does not impact our estimates of R^* (Supplement S7), the previously described factors nevertheless demonstrate that the impacts of heterogeneity between hosts on population-scale transmission should be explored further."*

4. **Comment:** I am wondering if there are any data or studies showing that the app usage and adherence decrease with increased number of notifications. For each individual, the probability of effective use may not depend on the notification time window and probably is more related to the perception of risk. On the other hand, if a person is notified about an exposure occurred 5 days ago, his/her adherence may be lower than when notified about an exposure 2 days ago. In any case, having some ballpark estimates of the decrease rate could inform whether a 5-day or 2-day window is more effective.

Response: We have now included a discussion of relevant available data from the UK and their implications. A direct evaluation of the impact of notification windows is difficult, because of confounding issues such as behavioural changes and other changes in restrictions occurring during the time. We note that the number of notified individuals in the UK per index case decreased after the rule change - however, this was already decreasing prior to the rule change. We also note that the number of positive tests after the rule change continued to decrease in England, tentatively suggesting that such a change may not have impacted impact users' likelihood of engagement with the app.

Added text (lines 114-124): *"For the NHS COVID-19 app, there are publicly available data regarding the number of notifications for each case reported on the app (NHS, 2021). The number of positive tests inputted into the app are also recorded, and can be combined with case data from UKHSA (GOV.UK, 2022) to infer the proportion of all known positive tests that are reported on the app (Kendall, 2022). These data may provide an insight into the impact of app rule changes. However, these data will also be impacted by behavioural trends at the time. For example, while the number of notified individuals per case reported on the app decreased after the notification window was reduced from five days to two days for asymptomatic individuals on the 2nd August 2021, it was already decreasing prior to the rule change. The proportion of all positive tests reported on the app decreased both before and after the rule change. This trend tentatively suggests that the change in the notification window duration had a limited impact on users' likelihood of engagement with the app; however, this decrease may have occurred more rapidly without the rule change."*

Reviewer 2

The manuscript "The effect of notification window length on the epidemiological impact of COVID-19 contact tracing mobile applications" uses a simple epidemiological model to investigate the effect of the notification window length for contact tracing apps on app usage effectiveness, in conjunction with public app usage and adherence to self-isolation. One important outcome is that, while increasing the length of the notification window allows to trace more contacts, it can be less effective of a shorter notification windows when combined with possible lower adherence.

I think the manuscript presents a highly valuable epidemiological study with important outcomes for public health, which is also strongly rooted in a concrete motivation – i.e., the change of notification policy in England and Wales in the summer 2021, and potential future decisions.

I find the paper well written, accessible to a wide readership in medical sciences and epidemiology. It presents a simple but rigorous method for evaluating effectiveness of contact tracing apps in terms of policy design (notification window) and public adherence (active app usage). The results are novel and robust, and the method looks scientifically sound.

I think one of the main messages of the paper – that a change in policy cannot be evaluated but in conjunction with the expected change in adherence – is extremely important for policy response, not only in the context of contact tracing but also for more

general public health interventions. Overall, I strongly recommend the paper for publication. I only have one main comment and several other minor comments that the authors may want to consider for a revision.

Main comments:

1a. Comment: A main comment regards the use of the quantity R^* defined as the ratio between average nr of secondary cases over primary cases. On the one hand, I do agree that counting primary cases only would neglect the effective contribution to transmission from those cases, which differs substantially according to the characteristics of the primary case. On the other hand, it seems to me that presenting R^* without an absolute measure of escaped transmission (i.e., presenting the average number of secondary cases per primary case, but without quantifying the actual number of primary cases) may not completely capture the absolute amount of infectious potential that has escaped. While I would launch an open question to the authors whether a quantity like “escaped infectious potential relative to the infectious potential without tracing” (looking at primary cases only) may be an alternative reasonable measure of relative effectiveness of interventions, I would encourage the authors to expand the explanation about the reasons for the choice of R^* as currently defined.

Response: We have now included a more detailed justification of our choice of R^* in the main text. Our definition of R^* in the paper is independent of the actual number of primary cases, as the number of secondary cases is scaled by the number of primary cases. Taking into account the expected number of secondary infections from primary infections is integral to quantifying the impact of notification windows and tracing more generally, as the expected number of primary infections is not impacted by contact tracing.

Added text (lines 70-75): *“Considering the expected number of secondary cases arising from primary cases is essential to quantify the impacts of different notification windows, as the expected number of primary cases is not affected by contact tracing (specifically, the expected number of primary cases depends only on when the base case isolates, and not on the notification window; our focus is the number of onwards transmissions prevented from primary cases as a result of the choice of notification window).”*

1b. Comment: As additional minor comment, the current calculations of R^* ignore the possibility that a symptomatic primary case is tested and input the positive result into the app, leading to notification and self-isolation of secondary cases, but I think this would only introduce second order effects and would not change the main results of the paper.

Response: We focus on this case as asymptomatic individuals were the subject of the rule change in the UK in August 2021. Whether the base case is symptomatic or asymptomatic will only change results in so far as the distribution of detection times would change. We have made a new figure, Supplementary Figure S2, corresponding to a base case individual who tests positive upon symptom onset, using the incubation period distribution defined by Lauer et al. used in the paper. We find that

detecting a symptomatic individual at symptom onset results in a larger reduction in R^* than detecting an asymptomatic individual through a one-off test, for either LFTs or PCR tests.

Minor comments:

1. **Comment:** On p. 1, the symbol “d” is used to explain the concept of notification window, however “d” is later used to denote the day of base case testing, and the symbol “w” to denote the notification window. It may be less confusing to keep using “w” from the beginning.

Response: w is now used throughout when referring to notification windows

2. **Comment:** Caption Fig. 2, (b), sentence “under the assumption that all notified individuals adhere to isolation upon notification”: I propose to change to “under the assumption that all individuals are active app users” (i.e., they use the app and adhere upon notification)

Response: Thanks for this suggestion- this is now mentioned

3. **Comment:** The base case here is considered to be asymptomatic. I find nothing wrong with this assumption, as I expect this to be a pessimistic scenario compared to a symptomatic case who would self-isolate upon symptoms. It may however be worth to briefly mention this explicitly.

Response: The impact of the base case being symptomatic is now considered in Supplement S2.

4. **Comment:** Supplementary material: I found the assumption of i_notif and $i_sympt = 10.5$ unusual in a continuous-time framework. Maybe mention explicitly this is an approximation, as I think that it would be possible in principle to model the “ten full days” exactly in the implementation?

Response: We have now stated that this an approximation explicitly in Supplement S1

Comment: Section S1.2: I have some doubts on some equations, which may be typos or my own mistakes, but the authors may want to check:

- a. Eq (9): should the lower integral bound be zero rather than “1”?
- b. Two lines after (9): If isolation starts from the day of last contact (day t), should the outbound be “ $t+i_notif$ ” rather than “ $d+i_notif-t$ ” ? (with corresponding time since infection i_notif , as done in eq (8) ?) In Eq (10), this would result in the upper bound of the second integral to be “ i_notif ” – which is the time since infection after isolating for i_notif days from last contact – rather than “ $d-t+i_notif$ ”
- c. Same doubts I have for the subcase (d) and corresponding equation (11)
- d. All the previous points – if I haven’t made mistakes myself – should affect (13) and the implementation of the codes
- e. Eq (14) there is an extra comma in the subscript

Response: We thank the reviewer for spotting these errors. The code has now been edited to adjust for these changes. Making these changes has only a minor impact on results.

5. **Comment:** S1.3, first line “the ratio between the expected number of primary cases and the expected number of secondary cases”: I think the sentence should be reversed
Response: This has now been amended
6. **Comment:** S1.4, first line: change “that” to “when”
Response: This has now been amended
7. **Comment:** Eq (17) and (18): You may want to use the inequality “>” rather than equality?
Response: This has now been amended

Reviewer 3

I reviewed this manuscript from an epidemiological point of view. I am not a mathematical modeller.

1. **Comment:** The authors explain notification windows for asymptomatic and symptomatic base cases (which are defined differently), and state that the UK policy change from 5 days to 2 days in response to the “pingdemic” only applied to asymptomatic cases. They included both asymptomatic and symptomatic base cases in their calculations (assuming that 30% of base cases were asymptomatic), but it was not clear to me whether they applied the notification window shift from 5 days to 2 days to asymptomatics only or also to symptomatics.
Response: Our main analysis focuses on the shift from 5 days to 2 days for asymptomatic individuals, which was the target of the rule change in England in August 2021. In the supplement, we now include a supplementary analysis considering the impact of assuming a base case is symptomatic, and assuming that the rule change also applies to symptomatic individuals.
2. **Comment:** The authors say that the success of contact-tracing apps depends on the proportion of infectious contacts notified (or at least I think that this is what they mean?) and the probability that those contacts quarantine after notification. However, instead of referring to the proportion of infectious contacts, they say number of contacts, number of infected individuals, number of infectious contacts, etc. in various places in the manuscript. I think what matters here is the proportion of all contacts, not the absolute numbers.
Response: We thank the reviewer for highlighting this needs clarification. In appropriate places in the text, ‘number’ has been replaced by ‘proportion’ - as the reviewer rightly suggests, it is the proportion rather than the number which is key to the success of contact tracing apps. We continue to use the term number when it is part of the phrase ‘expected number’, which is the formal probabilistic term. Much of the analysis is based upon estimates of the expected number of primary and secondary cases, hence in these instances we continue to refer to quantities in terms of numbers rather than proportions.
3. **Comment:** Related to this, the authors assume that the contacts occur randomly at a constant rate until taking a test. I could not find any information on the assumed rate,

and whether that rate was assumed to be equal for each base case. Contact matrices show that the contact rate differs substantially by age group and other factors. I realise that the authors had to simplify things but the reader needs to know what exactly they assumed regarding contact rates.

Response: We thank the reviewer for highlighting that our assumptions on assumed contact rates need clarification. No specific rate is specified because the results of the paper are not impacted by the specific rates, as higher contact rates scale both primary and secondary cases by the same value. This is now stated explicitly in the Supplement S1.

Added text (lines 351-353): *"While R is a function of an individual's contact rate, we find that R^* under different length notification windows is a linear function of R . Consequently, the relative effectiveness of different notification windows is independent of R , i.e. is independent of the contact rate assumed."*

Further response: Assuming contact rates are constant throughout an individual's infectious period, heterogeneity in contact rates between individuals does not affect the results of the paper. This is now demonstrated in Supplementary Figure S6, where the contact rates of base case and primary cases are now drawn from a lognormal distribution (0, 1.1).

- Comment:** I also wondered what the authors assumed regarding the sensitivity of the lateral flow test. One of the conclusions in the supplement is that R is less reduced by PCR testing than by lateral flow testing because of the delay in receiving a test result, but people are supposed to quarantine until they receive a test result. Furthermore, lateral flow tests produce many false-negative results, especially in asymptomatics (most studies find sensitivities of only 50-60%) and therefore miss many more cases than PCR tests. Again, I realise that this may be too much detail for this paper but the authors should acknowledge this problem with their simplifications and refrain from drawing conclusions that are misleading.

Response: In the supplement we find that there is a lower reduction in R when a base case individual is detected via a PCR test because of the delay in receiving a result. We assume that such individuals quarantine until they receive their result. R has a lower reduction in this instance (compared to a base case being detected via a LFT) for two reasons. Firstly, because there is a delay between a base case taking a test and primary cases who they have infected being notified, those primary case infectious individuals can have a longer period to make contacts with others, with the potential for onward transmission of infection, before isolating before isolating. Secondly, as individuals are more likely to be detected later in their infectious period with a PCR test, it becomes more likely that primary cases are infected prior to the notification window. We have now clarified this in the supplement.

Added text (lines 477-483): *" R is reduced by a lower extent in this instance for two reasons. Firstly, there is a delay between the base case taking a test and primary cases who they have infected being notified. Because of this, primary case infectious*

individuals can have a longer period to make contacts with others, with the potential for onward transmission of infection, before isolating. Secondly, owing to the greater sensitivity of PCR tests, individuals are more likely to be detected late in their infectious period. For late detected individuals, the majority of primary cases would be expected to have been infected prior to the notification window.”

Further Response: The reviewer is correct that lateral flow tests are less sensitive than PCR tests. Our results are conditionally predicated on individuals testing positive to a test. Our results should not be taken as a quantification of the relative merits of LFT and PCR testing. Rather, our results should be interpreted as: if an individual is detected by an LFT, contact tracing will lead to a greater reduction in transmission than if they were detected by a PCR. This is now stated explicitly in the supplementary information.

Further added text (lines 487-492): *“The results in Figure 2 and Figure S1 are predicated on the assumption that a base case individual tests positive to an LFT or a PCR respectively. However, LFTs are less sensitive than PCR tests (Pickering et al. 2021). Because of this, our results should not be taken as a quantification of the relative merits of LFT and PCR testing. Rather, our results should be interpreted as: if an individual is detected by an LFT and can input their result into the app immediately, contact tracing will lead to a greater reduction in transmission than if they were detected by a PCR and must wait until the PCR result has returned.”*

5. **Comment:** The authors do not compare and contrast their findings with those by other mathematical modelers. They also do not compare their calculations with real-life data from the UK before and after the policy change on 2 August 2021. These comparisons are important precisely because their calculations are simplified versions of reality.

Response: We have now included a longer discussion of other modelling work and relevant UK data before and after the policy change on 2 August 2021. The discussion of relevant data is included earlier in this document. The following text has been added discussing relevant modelling work:

Added text (lines 125-136): *“Previous modelling studies have demonstrated that the extent of active usage of mobile contact tracing apps has a large impact on their effectiveness (Wymant et al. 2021). Time to notification of exposed contacts also plays an integral role for SARS-CoV-2 (Ferretti et al. 2020) ; this is corroborated by our results regarding the contrasting reduction in R^* based on whether a user inputs their positive result from an LFT or from a PCR test (where the PCR test involves a delay between the test date and the positive test result). A range of other factors may also impact the effectiveness of app-based measures. For example, a policy that increases the duration of self-isolation may increase the risk of transmission, if such a policy also leads to a decrease in the rate of self-reporting and adherence (Lucas et al. 2021). It has been suggested that contact tracing has the potential to delay epidemics, in part because of significant heterogeneity in transmission of SARS-CoV-2 between different infected individuals (Gardner and Fitzpatrick, 2021). While we find that heterogeneity in contact rates or duration of infection does not impact our estimates of R^* (Supplement*

S7), the previously described factors nevertheless demonstrate that the impacts of heterogeneity between hosts on population-scale transmission should be explored further. ”

6. **Comment:** Finally, I was wondering whether another way out of the “pingdemic” could have been to change the definition of close contact from 2 meters to 1.5 meters (as is the case in many other countries).

Response: The reviewer is correct that reducing the definition of a close contact according to proximity would be another option to reduce the number of contacts notified per identified case. This could be incorporated into our framework by reducing the probability that individual’s who fall within the notification window are notified, much in the same way as active app use is implemented in the model. We have included a discussion of relevant prior modelling work alongside a discussion of the value of considering the definition of a close contact in future studies:

Added text (lines 138-151): *“While our study considers the impact of reducing the number of notified individuals by shortening the length of the notification window, a reduction in the number of notified individuals could also be achieved by changing the definition of a high-risk encounter. While the current definition involves being within two metres of someone for at least fifteen minutes (NHS, 2021), this threshold distance could be decreased or the threshold duration of contact increased. Sensitivity of our results to either of these factors could be approximated in our framework by reducing the probability of primary cases being notified via the app. Though there are relatively few studies considering the impact of mobile contact tracing apps specifically, studies exploring the impact of contact tracing more generally have considered the impact of reducing the proximity or increasing the duration of contact for someone to be regarded as a “close contact” for SARS-CoV-2. The impact of duration has been considered explicitly (Keeling et al. 2020), while other studies have explored the impact of reducing the overall percentage of contacts traced (Kretzschmar et al. 2020, Hellewell et al. 2020), including through varying the notification window (Hart et al. 2021). Despite practical limitations preventing precise measurements of both proximity and duration of contact through mobile devices (Kindt et al. 2021), future studies exploring the effects of these factors on transmission would be a valuable line of research.”*

REVIEWERS' COMMENTS:

Reviewer #1 (Remarks to the Author):

I appreciate the authors' efforts in revising the manuscript. The additional analyses and clarifications have addressed my questions. I also evaluated the responses to the comments of Reviewer 3 and found that the questions were properly answered.

Sen Pei

Reviewer #2 (Remarks to the Author):

I thank the authors for clarifying all of my previous doubts and addressing all my comments. I only have a couple of minor comments below.

P 5, line 105: receives -> receive

P 5, l 124: more rapidly? Or slowly?

P 6, l 171-173 may suggest that symptomatic and asymptomatic have same infectiousness, although I understand from P10, l 384-385 that asymptomatic infectiousness is also scaled (as for vaccinated)

P16 l 469: extra "is"

P17 l 481-482 "owing to the greater sensitivity of PCR tests, individuals are more likely to be detected late in their infectious period." – I'm slightly confused as why this should contribute to reduce effectiveness of contact tracing: owing to the greater sensitivity of PCR, I would expect these cases to be detected IN ADDITION TO the cases detected by LFT, so although they are detected late in their infectious period, they should still contribute positively to contact tracing.

P 23: typo "[!ht]" on top of the page

Response to reviewers 2: The effect of notification window length on the epidemiological impact of COVID-19 contact tracing mobile applications

We thank both reviewers for their valuable comments. Reviewers' comments are in blue text, our responses are in black text, while relevant sections of added text are included in italics. The specific part of a reviewer's comment that we are responding to is in bold blue font before the corresponding response.

Reviewer 1:

I appreciate the authors' efforts in revising the manuscript. The additional analyses and clarifications have addressed my questions. I also evaluated the responses to the comments of Reviewer 3 and found that the questions were properly answered.

Reviewer 2:

I thank the authors for clarifying all of my previous doubts and addressing all my comments. I only have a couple of minor comments below.

1. Comment: **P 5, line 105: receives -> receive**
Response: this has now been amended.
2. Comment: **P 5, I 124: more rapidly? Or slowly?**
Response: As the motivation for the app rule change on 2nd of August 2021 was to encourage the continued use of the app, we speculate that without the rule change the number of people actively engaging with the app may have decreased more rapidly. Hence, we have kept our original wording.
3. Comment: **P 6, I 171-173 may suggest that symptomatic and asymptomatic have same infectiousness, although I understand from P10, I 384-385 that asymptomatic infectiousness is also scaled (as for vaccinated)**
Response: Thanks for noticing this. We have amended our text in the discussion to clarify that asymptomatic infectiousness is also scaled.
Amended text (added text in bold): *"The infectiousness profiles of symptomatic, asymptomatic, and vaccinated individuals are assumed to be equal in this study (at least up to the time at which symptomatic individuals develop symptoms, at which point they isolate), though the profiles of asymptomatic and vaccinated individuals are scaled so that they are expected to generate fewer infections overall."*
4. Comment: **P16 I 469: extra "is"**
Response: This has now been removed.
5. Comment: **P17 I 481-482 "owing to the greater sensitivity of PCR tests, individuals are more likely to be detected late in their infectious period." – I'm slightly confused as why this should contribute to reduce effectiveness of contact tracing: owing to the greater sensitivity of PCR, I would expect these cases to be detected IN ADDITION TO the cases detected by LFT, so although they are detected late in their infectious period, they should still contribute positively to contact tracing.**
Response: We thank the reviewer for highlighting this. We now state this point explicitly in the Supplementary Text 2:
Added text: *"It is important to note that detection via PCR tests still contributes positively to contact tracing, particularly if PCR tests detect individuals who test negative to an LFT."*
6. Comment: **P 23: typo "[!ht]" on top of the page**
Response: This has now been removed.